# On the Number of Linear Regions of Deep Neural Networks

**Guido Montúfar**
Max Planck Institute for Mathematics in the Sciences
montufar@mis.mpg.de

**Razvan Pascanu**
Université de Montréal
pascanur@iro.umontreal.ca

**Kyunghyun Cho**
Université de Montréal
kyunghyun.cho@umontreal.ca

**Yoshua Bengio**
Université de Montréal, CIFAR Fellow
yoshua.bengio@umontreal.ca

## Abstract

We study the complexity of functions computable by deep feedforward neural networks with piecewise linear activations in terms of the symmetries and the number of linear regions that they have. Deep networks are able to sequentially map portions of each layer's input-space to the same output. In this way, deep models compute functions that react equally to complicated patterns of different inputs. The compositional structure of these functions enables them to re-use pieces of computation exponentially often in terms of the network's depth. This paper investigates the complexity of such compositional maps and contributes new theoretical results regarding the advantage of depth for neural networks with piecewise linear activation functions. In particular, our analysis is not specific to a single family of models, and as an example, we employ it for rectifier and maxout networks. We improve complexity bounds from pre-existing work and investigate the behavior of units in higher layers.

**Keywords:** Deep learning, neural network, input space partition, rectifier, maxout

## 1 Introduction

Artificial neural networks with several hidden layers, called *deep* neural networks, have become popular due to their unprecedented success in a variety of machine learning tasks (see, e.g., Krizhevsky et al. 2012, Ciresan et al. 2012, Goodfellow et al. 2013, Hinton et al. 2012). In view of this empirical evidence, deep neural networks are becoming increasingly favored over *shallow* networks (i.e., with a single layer of hidden units), and are often implemented with more than five layers. At the time being, however, the theory of deep networks still poses many questions. Recently, Delalleau and Bengio (2011) showed that a shallow network requires exponentially many more sum-product hidden units[1] than a deep sum-product network in order to compute certain families of polynomials. We are interested in extending this kind of analysis to more popular neural networks, such as those with maxout and rectifier units.

There is a wealth of literature discussing approximation, estimation, and complexity of artificial neural networks (see, e.g., Anthony and Bartlett 1999). A well-known result states that a feedforward neural network with a single, huge, hidden layer is a universal approximator of Borel measurable functions (see Hornik et al. 1989, Cybenko 1989). Other works have investigated universal approximation of probability distributions by deep belief networks (Le Roux and Bengio 2010, Montúfar and Ay 2011), as well as their approximation properties (Montúfar 2014, Krause et al. 2013).

These previous theoretical results, however, do not trivially apply to the types of deep neural networks that have seen success in recent years. Conventional neural networks often employ either hidden units

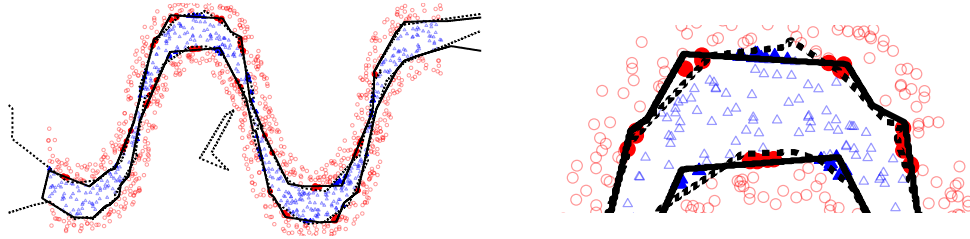

Figure 1: Binary classification using a shallow model with 20 hidden units (solid line) and a deep model with two layers of 10 units each (dashed line). The right panel shows a close-up of the left panel. Filled markers indicate errors made by the shallow model.

with a bounded smooth activation function, or Boolean hidden units. On the other hand, recently it has become more common to use piecewise linear functions, such as the *rectifier* activation $g(a) = \max\{0, a\}$ (Glorot et al. 2011, Nair and Hinton 2010) or the *maxout* activation $g(a_1, \ldots, a_k) = \max\{a_1, \ldots, a_k\}$ (Goodfellow et al. 2013). The practical success of deep neural networks with piecewise linear units calls for the theoretical analysis specific for this type of neural networks.

In this respect, Pascanu et al. (2013) reported a theoretical result on the complexity of functions computable by deep feedforward networks with rectifier units. They showed that, in the asymptotic limit of many hidden layers, deep networks are able to separate their input space into exponentially more linear response regions than their shallow counterparts, despite using the same number of computational units.

Building on the ideas from Pascanu et al. (2013), we develop a general framework for analyzing deep models with piecewise linear activations. We describe how the intermediary layers of these models are able to map several pieces of their inputs into the same output. The layer-wise composition of the functions computed in this way re-uses low-level computations exponentially often as the number of layers increases. This key property enables deep networks to compute highly complex and structured functions. We underpin this idea by estimating the number of linear regions of functions computable by two important types of piecewise linear networks: with rectifier units and with maxout units. Our results for the complexity of deep rectifier networks yield a significant improvement over the previous results on rectifier networks mentioned above, showing a favorable behavior of deep over shallow networks even with a moderate number of hidden layers. Furthermore, our analysis of deep rectifier and maxout networks provides a platform to study a broad variety of related networks, such as convolutional networks.

The number of linear regions of the functions that can be computed by a given model is a measure of the model's flexibility. An example of this is given in Fig. 1, which compares the learned decision boundary of a single-layer and a two-layer model with the same number of hidden units (see details in the Supplementary Material). This illustrates the advantage of depth; the deep model captures the desired boundary more accurately, approximating it with a larger number of linear pieces. As noted earlier, deep networks are able to *identify* an exponential number of input neighborhoods by mapping them to a common output of some intermediary hidden layer. The computations carried out on the activations of this intermediary layer are replicated many times, once in each of the identified neighborhoods. This allows the networks to compute very complex looking functions even when they are defined with relatively few parameters. The number of parameters is an upper bound for the dimension of the set of functions computable by a network, and a small number of parameters means that the class of computable functions has a low dimension. The set of functions computable by a deep feedforward piecewise linear network, although low dimensional, achieves exponential complexity by re-using and composing features from layer to layer.

## 2 Feedforward Neural Networks and their Compositional Properties

In this section we discuss the ability of deep feedforward networks to re-map their input-space to create complex symmetries by using only relatively few computational units. The key observation of our analysis is that each layer of a deep model is able to map different regions of its input to a common output. This leads to a compositional structure, where computations on higher layers are effectively replicated in all input regions that produced the same output at a given layer. The capacity to replicate computations over the input-space grows exponentially with the number of network layers. Before expanding these ideas, we introduce basic definitions needed in the rest of the paper. At the end of this section, we give an intuitive perspective for reasoning about the replicative capacity of deep models.

## 2.1 Definitions

A *feedforward neural network* is a composition of layers of computational units which defines a function $F \colon \mathbb{R}^{n_0} \to \mathbb{R}^{\mathrm{out}}$ of the form

$$F(\mathbf{x}; \theta) = f_{\mathrm{out}} \circ g_L \circ f_L \circ \cdots \circ g_1 \circ f_1(\mathbf{x}), \tag{1}$$

where $f_l$ is a linear preactivation function and $g_l$ is a nonlinear activation function. The parameter $\theta$ is composed of *input* weight matrices $\mathbf{W}_l \in \mathbb{R}^{k \cdot n_l \times n_{l-1}}$ and *bias* vectors $\mathbf{b}_l \in \mathbb{R}^{k \cdot n_l}$ for each layer $l \in [L]$.

The output of the $l$-th layer is a vector $\mathbf{x}_l = [\mathbf{x}_{l,1}, \ldots, \mathbf{x}_{l,n_l}]^\top$ of activations $\mathbf{x}_{l,i}$ of the units $i \in [n_l]$ in that layer. This is computed from the activations of the preceding layer by $\mathbf{x}_l = g_l(f_l(\mathbf{x}_{l-1}))$. Given the activations $\mathbf{x}_{l-1}$ of the units in the $(l-1)$-th layer, the preactivation of layer $l$ is given by

$$f_l(\mathbf{x}_{l-1}) = \mathbf{W}_l \mathbf{x}_{l-1} + \mathbf{b}_l,$$

where $f_l = [f_{l,1}, \ldots, f_{l,n_l}]^\top$ is an array composed of $n_l$ preactivation vectors $f_{l,i} \in \mathbb{R}^k$, and the activation of the $i$-th unit in the $l$-th layer is given by

$$\mathbf{x}_{l,i} = g_{l,i}(f_{l,i}(\mathbf{x}_{l-1})).$$

We will abbreviate $g_l \circ f_l$ by $h_l$. When the layer index $l$ is clear, we will drop the corresponding subscript. We are interested in piecewise linear activations, and will consider the following two important types.

- Rectifier unit: $\quad g_i(f_i) = \max\{0, f_i\}$, where $f_i \in \mathbb{R}$ and $k = 1$.
- Rank-$k$ maxout unit: $\ g_i(f_i) = \max\{f_{i,1}, \ldots, f_{i,k}\}$, where $f_i = [f_{i,1}, \ldots, f_{i,k}] \in \mathbb{R}^k$.

The *structure* of the network refers to the way its units are arranged. It is specified by the number $n_0$ of input dimensions, the number of layers $L$, and the number of units or *width* $n_l$ of each layer.

We will classify the functions computed by different network structures, for different choices of parameters, in terms of their number of linear regions. A *linear region* of a piecewise linear function $F \colon \mathbb{R}^{n_0} \to \mathbb{R}^m$ is a maximal connected subset of the input-space $\mathbb{R}^{n_0}$, on which $F$ is linear. For the functions that we consider, each linear region has full dimension, $n_0$.

## 2.2 Shallow Neural Networks

Rectifier units have two types of behavior; they can be either constant 0 or linear, depending on their inputs. The boundary between these two behaviors is given by a hyperplane, and the collection of all the hyperplanes coming from all units in a rectifier layer forms a *hyperplane arrangement*. In general, if the activation function $g \colon \mathbb{R} \to \mathbb{R}$ has a distinguished (i.e., irregular) behavior at zero (e.g., an inflection point or non-linearity), then the function $\mathbb{R}^{n_0} \to \mathbb{R}^{n_1}$; $\mathbf{x} \mapsto g(\mathbf{W}\mathbf{x} + \mathbf{b})$ has a distinguished behavior at all inputs from any of the hyperplanes $H_i := \{\mathbf{x} \in \mathbb{R}^{n_0} \colon \mathbf{W}_{i,:}\mathbf{x} + \mathbf{b}_i = 0\}$ for $i \in [n_1]$. The hyperplanes capturing this distinguished behavior also form a hyperplane arrangement (see, e.g., Pascanu et al. 2013).

The hyperplanes in the arrangement split the input-space into several regions. Formally, a *region* of a hyperplane arrangement $\{H_1, \ldots, H_{n_1}\}$ is a connected component of the complement $\mathbb{R}^{n_0} \setminus (\cup_i H_i)$, i.e., a set of points delimited by these hyperplanes (possibly open towards infinity). The number of regions of an arrangement can be given in terms of a characteristic function of the arrangement, as shown in a well-known result by Zaslavsky (1975). An arrangement of $n_1$ hyperplanes in $\mathbb{R}^{n_0}$ has at most $\sum_{j=0}^{n_0} \binom{n_1}{j}$ regions. Furthermore, this number of regions is attained if and only if the hyperplanes are in general position. This implies that the maximal number of linear regions of functions computed by a shallow rectifier network with $n_0$ inputs and $n_1$ hidden units is $\sum_{j=0}^{n_0} \binom{n_1}{j}$ (see Pascanu et al. 2013; Proposition 5).

## 2.3 Deep Neural Networks

We start by defining the identification of input neighborhoods mentioned in the introduction more formally:

**Definition 1.** A map $F$ *identifies* two neighborhoods $S$ and $T$ of its input domain if it maps them to a common subset $F(S) = F(T)$ of its output domain. In this case we also say that $S$ and $T$ are *identified* by $F$.

**Example 2.** The four quadrants of 2-D Euclidean space are regions that are identified by the absolute value function $g \colon \mathbb{R}^2 \to \mathbb{R}^2$; $(x_1, x_2) \mapsto [|x_1|, |x_2|]^\top$.

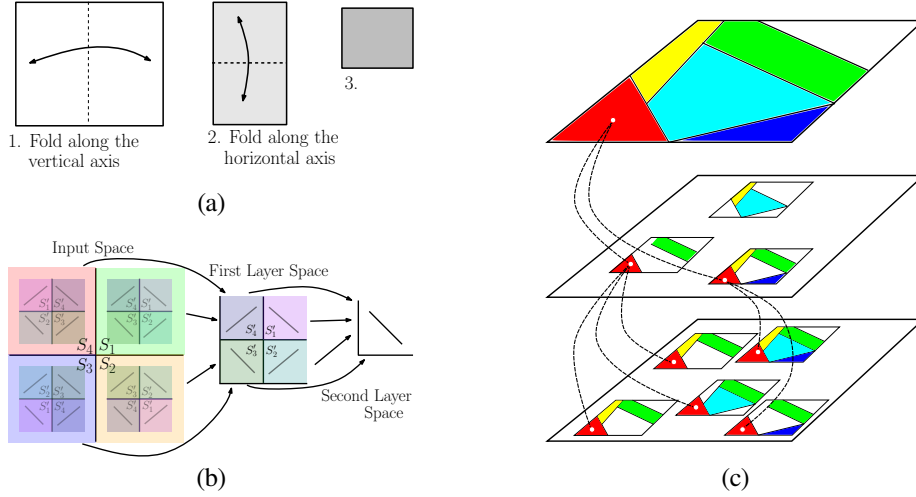

Figure 2: (a) Space folding of 2-D Euclidean space along the two coordinate axes. (b) An illustration of how the top-level partitioning (on the right) is replicated to the original input space (left). (c) Identification of regions across the layers of a deep model.

The computation carried out by the $l$-th layer of a feedforward network on a set of activations from the $(l-1)$-th layer is effectively carried out for all regions of the input space that lead to the same activations of the $(l-1)$-th layer. One can choose the input weights and biases of a given layer in such a way that the computed function behaves most interestingly on those activation values of the preceding layer which have the largest number of preimages in the input space, thus replicating the interesting computation many times in the input space and generating an overall complicated-looking function.

For any given choice of the network parameters, each hidden layer $l$ computes a function $h_l = g_l \circ f_l$ on the output activations of the preceding layer. We consider the function $F_l \colon \mathbb{R}^{n_0} \to \mathbb{R}^{n_l}$; $F_l := h_l \circ \cdots \circ h_1$ that computes the activations of the $l$-th hidden layer. We denote the image of $F_l$ by $S_l \subseteq \mathbb{R}^{n_l}$, i.e., the set of (vector valued) activations reachable by the $l$-th layer for all possible inputs. Given a subset $R \subseteq S_l$, we denote by $P_R^l$ the set of subsets $\bar{R}_1, \ldots, \bar{R}_k \subseteq S_{l-1}$ that are mapped by $h_l$ onto $R$; that is, subsets that satisfy $h_l(\bar{R}_1) = \cdots = h_l(\bar{R}_k) = R$. See Fig. 2 for an illustration.

The number of separate input-space neighborhoods that are mapped to a common neighborhood $R \subseteq S_l \subseteq \mathbb{R}^{n_l}$ can be given recursively as

$$\mathcal{N}_R^l = \sum_{R' \in P_R^l} \mathcal{N}_{R'}^{l-1}, \qquad \mathcal{N}_R^0 = 1, \text{ for each region } R \subseteq \mathbb{R}^{n_0}. \tag{2}$$

For example, $P_R^1$ is the set of all disjoint input-space neighborhoods whose image by the function computed by the first layer, $h_1 \colon \mathbf{x} \mapsto g(\mathbf{W}\mathbf{x} + \mathbf{b})$, equals $R \subseteq S_1 \subseteq \mathbb{R}^{n_1}$.

The recursive formula (2) counts the number of identified sets by moving along the branches of a tree rooted at the set $R$ of the $j$-th layer's output-space (see Fig. 2 (c)). Based on these observations, we can estimate the maximal number of linear regions as follows.

**Lemma 3.** *The maximal number of linear regions of the functions computed by an $L$-layer neural network with piecewise linear activations is at least $\mathcal{N} = \sum_{R \in P^L} \mathcal{N}_R^{L-1}$, where $\mathcal{N}_R^{L-1}$ is defined by Eq. (2), and $P^L$ is a set of neighborhoods in distinct linear regions of the function computed by the last hidden layer.*

Here, the idea to construct a function with many linear regions is to use the first $L-1$ hidden layers to identify many input-space neighborhoods, mapping all of them to the activation neighborhoods $P^L$ of the $(L-1)$-th hidden layer, each of which belongs to a distinct linear region of the last hidden layer. We will follow this strategy in Secs. 3 and 4, where we analyze rectifier and maxout networks in detail.

## 2.4 Identification of Inputs as Space Foldings

In this section, we discuss an intuition behind Lemma 3 in terms of *space folding*. A map $F$ that identifies two subsets $\mathcal{S}$ and $\mathcal{S}'$ can be considered as an operator that *folds* its domain in such a way that the two

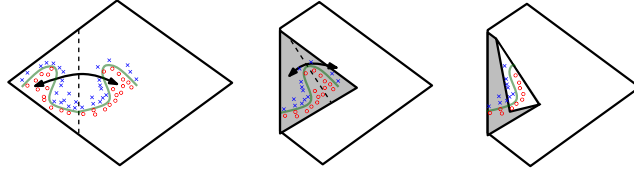

Figure 3: Space folding of 2-D space in a non-trivial way. Note how the folding can potentially identify symmetries in the boundary that it needs to learn.

subsets $\mathcal{S}$ and $\mathcal{S}'$ coincide and are mapped to the same output. For instance, the absolute value function $g \colon \mathbb{R}^2 \to \mathbb{R}^2$ from Example 2 folds its domain twice (once along each coordinate axis), as illustrated in Fig. 2 (a). This folding identifies the four quadrants of 2-D Euclidean space. By composing such operations, the same kind of map can be applied again to the output, in order to re-fold the first folding.

Each hidden layer of a deep neural network can be associated with a folding operator. Each hidden layer folds the space of activations of the previous layer. In turn, a deep neural network effectively folds its input-space recursively, starting with the first layer. The consequence of this recursive folding is that any function computed on the final folded space will apply to all the collapsed subsets identified by the map corresponding to the succession of foldings. This means that in a deep model any partitioning of the last layer's image-space is replicated in all input-space regions which are identified by the succession of foldings. Fig. 2 (b) offers an illustration of this replication property.

Space foldings are not restricted to foldings along coordinate axes and they do not have to preserve lengths. Instead, the space is folded depending on the orientations and shifts encoded in the input weights $\mathbf{W}$ and biases $\mathbf{b}$ and on the nonlinear activation function used at each hidden layer. In particular, this means that the sizes and orientations of identified input-space regions may differ from each other. See Fig. 3. In the case of activation functions which are not piece-wise linear, the folding operations may be even more complex.

### 2.5 Stability to Perturbation

Our bounds on the complexity attainable by deep models (Secs. 3 and 4) are based on suitable choices of the network weights. However, this does not mean that the indicated complexity is only attainable in singular cases. The parametrization of the functions computed by a neural network is continuous. More precisely, the map $\psi \colon \mathbb{R}^N \to C(\mathbb{R}^{n_0}; \mathbb{R}^{n_L}); \; \theta \mapsto F_\theta$, which maps input weights and biases $\theta = \{\mathbf{W}_i, \mathbf{b}_i\}_{i=1}^{L}$ to the continuous functions $F_\theta \colon \mathbb{R}^{n_0} \to \mathbb{R}^{n_L}$ computed by the network, is continuous. Our analysis considers the number of linear regions of the functions $F_\theta$. By definition, each linear region contains an open neighborhood of the input-space $\mathbb{R}^{n_0}$. Given any function $F_\theta$ with a finite number of linear regions, there is an $\epsilon > 0$ such that for each $\epsilon$-perturbation of the parameter $\theta$, the resulting function $F_{\theta+\epsilon}$ has at least as many linear regions as $F_\theta$. The linear regions of $F_\theta$ are preserved under small perturbations of the parameters, because they have a finite volume.

If we define a probability density on the space of parameters, what is the probability of the event that the function represented by the network has a given number of linear regions? By the above discussion, the probability of getting a number of regions at least as large as the number resulting from any particular choice of parameters (for a uniform measure within a bounded domain) is nonzero, even though it may be very small. This is because there exists an epsilon-ball of non-zero volume around that particular choice of parameters, for which at least the same number of linear regions is attained. For example, shallow rectifier networks generically attain the maximal number of regions, even if in close vicinity of any parameter choice there may be parameters corresponding to functions with very few regions.

For future work it would be interesting to study the partitions of parameter space $\mathbb{R}^N$ into pieces where the resulting functions partition their input-spaces into isomorphic linear regions, and to investigate how many of these pieces of parameter space correspond to functions with a given number of linear regions.

### 2.6 Empirical Evaluation of Folding in Rectifier MLPs

We empirically examined the behavior of a trained MLP to see if it folds the input-space in the way described above. First, we note that tracing the activation of each hidden unit in this model gives a piecewise linear map $\mathbb{R}^{n_0} \to \mathbb{R}$ (from inputs to activation values of that unit). Hence, we can analyze the behavior of each

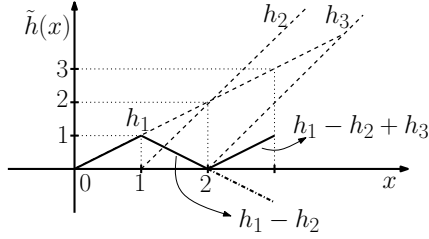

Figure 4: Folding of the real line into equal-length segments by a sum of rectifiers.

unit by visualizing the different weight matrices corresponding to the different linear pieces of this map. The weight matrix of one piece of this map can be found by tracking the linear piece used in each intermediary layer, starting from an input example. This visualization technique, a byproduct of our theoretical analysis, is similar to the one proposed by Zeiler and Fergus (2013), but is motivated by a different perspective.

After computing the activations of an intermediary hidden unit for each training example, we can, for instance, inspect two examples that result in similar levels of activation for a hidden unit. With the linear maps of the hidden unit corresponding to the two examples we perturb one of the examples until it results in exactly the same activation. These two inputs then can be safely considered as points in two regions identified by the hidden unit. In the Supplementary Material we provide details and examples of this visualization technique. We also show inputs identified by a deep MLP.

## 3   Deep Rectifier Networks

In this section we analyze deep neural networks with rectifier units, based on the general observations from Sec. 2. We improve upon the results by Pascanu et al. (2013), with a tighter lower-bound on the maximal number of linear regions of functions computable by deep rectifier networks. First, let us note the following upper-bound, which follows directly from the fact that each linear region of a rectifier network corresponds to a pattern of hidden units being active:

**Proposition 4.** *The maximal number of linear regions of the functions computed by any rectifier network with a total of $N$ hidden units is bounded from above by $2^N$.*

### 3.1   Illustration of the Construction

Consider a layer of $n$ rectifiers with $n_0$ input variables, where $n \geq n_0$. We partition the set of rectifier units into $n_0$ (non-overlapping) subsets of cardinality $p = \lfloor n/n_0 \rfloor$ and ignore the remainder units. Consider the units in the $j$-th subset. We can choose their input weights and biases such that

$$
\begin{aligned}
h_1(\mathbf{x}) &= \max\{0,\ \mathbf{wx}\}, \\
h_2(\mathbf{x}) &= \max\{0, 2\mathbf{wx} - 1\}, \\
h_3(\mathbf{x}) &= \max\{0, 2\mathbf{wx} - 2\}, \\
&\quad\vdots \\
h_p(\mathbf{x}) &= \max\{0, 2\mathbf{wx} - (p-1)\},
\end{aligned}
$$

where $\mathbf{w}$ is a row vector with $j$-th entry equal to $1$ and all other entries set to $0$. The product $\mathbf{wx}$ selects the $j$-th coordinate of $\mathbf{x}$. Adding these rectifiers with alternating signs, we obtain following scalar function:

$$
\tilde{h}_j(\mathbf{x}) = \left[1, -1, 1, \ldots, (-1)^{p-1}\right] \left[h_1(\mathbf{x}), h_2(\mathbf{x}), h_3(\mathbf{x}), \ldots, h_p(\mathbf{x})\right]^\top. \tag{3}
$$

Since $\tilde{h}_j$ acts only on the $j$-th input coordinate, we may redefine it to take a scalar input, namely the $j$-th coordinate of $\mathbf{x}$. This function has $p$ linear regions given by the intervals $(-\infty, 0]$, $[0, 1]$, $[1, 2]$, $\ldots$, $[p-1, \infty)$. Each of these intervals has a subset that is mapped by $\tilde{h}_j$ onto the interval $(0, 1)$, as illustrated in Fig. 4. The function $\tilde{h}_j$ identifies the input-space strips with $j$-th coordinate $\mathbf{x}_j$ restricted to the intervals $(0, 1), (1, 2), \ldots, (p-1, p)$. Consider now all the $n_0$ subsets of rectifiers and the function $\tilde{h} = \left[\tilde{h}_1, \tilde{h}_2, \ldots, \tilde{h}_p\right]^\top$. This function is locally symmetric about each hyperplane with a fixed $j$-th coordinate

equal to $\mathbf{x}_j = 1, \ldots, \mathbf{x}_j = p - 1$ (vertical lines in Fig. 4), for all $j = 1, \ldots, n_0$. Note the periodic pattern that emerges. In fact, the function $\tilde{h}$ identifies a total of $p^{n_0}$ hypercubes delimited by these hyperplanes.

Now, note that $\tilde{h}$ arises from $h$ by composition with a linear function (alternating sums). This linear function can be effectively absorbed in the preactivation function of the next layer. Hence we can treat $\tilde{h}$ as being the function computed by the current layer. Computations by deeper layers, as functions of the unit hypercube output of this rectifier layer, are replicated on each of the $p^{n_0}$ identified input-space hypercubes.

## 3.2 Formal Result

We can generalize the construction described above to the case of a deep rectifier network with $n_0$ inputs and $L$ hidden layers of widths $n_i \geq n_0$ for all $i \in [L]$. We obtain the following lower bound for the maximal number of linear regions of deep rectifier networks:

**Theorem 5.** *The maximal number of linear regions of the functions computed by a neural network with $n_0$ input units and $L$ hidden layers, with $n_i \geq n_0$ rectifiers at the $i$-th layer, is lower bounded by*

$$\left( \prod_{i=1}^{L-1} \left\lfloor \frac{n_i}{n_0} \right\rfloor^{n_0} \right) \sum_{j=0}^{n_0} \binom{n_L}{j}.$$

The next corollary gives an expression for the asymptotic behavior of these bounds. Assuming that $n_0 = O(1)$ and $n_i = n$ for all $i \geq 1$, the number of regions of a single layer model with $Ln$ hidden units behaves as $O(L^{n_0} n^{n_0})$ (see Pascanu et al. 2013; Proposition 10). For a deep model, Theorem 5 implies:

**Corollary 6.** *A rectifier neural network with $n_0$ input units and $L$ hidden layers of width $n \geq n_0$ can compute functions that have $\Omega\left( \left( n/n_0 \right)^{(L-1)n_0} n^{n_0} \right)$ linear regions.*

Thus we see that the number of linear regions of deep models grows exponentially in $L$ and polynomially in $n$, which is much faster than that of shallow models with $nL$ hidden units. Our result is a significant improvement over the bound $\Omega\left( \left( n/n_0 \right)^{L-1} n^{n_0} \right)$ obtained by Pascanu et al. (2013). In particular, our result demonstrates that even for small values of $L$ and $n$, deep rectifier models are able to produce substantially more linear regions than shallow rectifier models. Additionally, using the same strategy as Pascanu et al. (2013), our result can be reformulated in terms of the number of *linear regions per parameter*. This results in a similar behavior, with deep models being exponentially more efficient than shallow models (see the Supplementary Material).

## 4 Deep Maxout Networks

A maxout network is a feedforward network with layers defined as follows:

**Definition 7.** A *rank-k maxout layer* with $n$ input and $m$ output units is defined by a preactivation function of the form $f: \mathbb{R}^n \to \mathbb{R}^{m \cdot k}$; $f(\mathbf{x}) = \mathbf{W}\mathbf{x} + \mathbf{b}$, with input and bias weights $\mathbf{W} \in \mathbb{R}^{m \cdot k \times n}$, $\mathbf{b} \in \mathbb{R}^{m \cdot k}$, and activations of the form $g_j(\mathbf{z}) = \max\{\mathbf{z}_{(j-1)k+1}, \ldots, \mathbf{z}_{jk}\}$ for all $j \in [m]$. The layer computes a function

$$g \circ f: \quad \mathbb{R}^n \to \mathbb{R}^m; \quad \mathbf{x} \mapsto \begin{pmatrix} \max\{f_1(\mathbf{x}), \ldots, f_k(\mathbf{x})\} \\ \vdots \\ \max\{f_{(m-1)k+1}(\mathbf{x}), \ldots, f_{mk}(\mathbf{x})\} \end{pmatrix}. \tag{4}$$

Since the maximum of two convex functions is convex, maxout units and maxout layers compute convex functions. The maximum of a collection of functions is called their *upper envelope*. We can view the graph of each linear function $f_i: \mathbb{R}^n \to \mathbb{R}$ as a supporting hyperplane of a convex set in $(n+1)$-dimensional space. In particular, if each $f_i$, $i \in [k]$ is the unique maximizer $f_i = \max\{f_{i'}': i' \in [k]\}$ at some input neighborhood, then the number of linear regions of the upper envelope $g_1 \circ f = \max\{f_i: i \in [k]\}$ is exactly $k$. This shows that the maximal number of linear regions of a maxout unit is equal to its rank.

The linear regions of the maxout layer are the intersections of the linear regions of the individual maxout units. In order to obtain the number of linear regions for the layer, we need to describe the structure of the linear regions of each maxout unit, and study their possible intersections. Voronoi diagrams can be

lifted to upper envelopes of linear functions, and hence they describe input-space partitions generated by maxout units. Now, how many regions do we obtain by intersecting the regions of $m$ Voronoi diagrams with $k$ regions each? Computing the intersections of Voronoi diagrams is not easy, in general. A trivial upper bound for the number of linear regions is $k^m$, which corresponds to the case where all intersections of regions of different units are different from each other. We will give a better bound in Proposition 8.

Now, for the purpose of computing lower bounds, here it will be sufficient to consider certain well-behaved special cases. One simple example is the division of input-space by $k-1$ parallel hyperplanes. If $m \leq n$, we can consider the arrangement of hyperplanes $H_i = \{\mathbf{x} \in \mathbb{R}^n : \mathbf{x}_j = i\}$ for $i = 1, \ldots, k-1$, for each maxout unit $j \in [m]$. In this case, the number of regions is $k^m$. If $m > n$, the same arguments yield $k^n$ regions.

**Proposition 8.** *The maximal number of regions of a single layer maxout network with $n$ inputs and $m$ outputs of rank $k$ is lower bounded by $k^{\min\{n,m\}}$ and upper bounded by $\min\{\sum_{j=0}^n \binom{k^2 m}{j}, k^m\}$.*

Now we take a look at the deep maxout model. Note that a rank-2 maxout layer can be simulated by a rectifier layer with twice as many units. Then, by the results from the last section, a rank-2 maxout network with $L-1$ hidden layers of width $n = n_0$ can identify $2^{n_0(L-1)}$ input-space regions, and, in turn, it can compute functions with $2^{n_0(L-1)}2^{n_0} = 2^{n_0 L}$ linear regions. For the rank-$k$ case, we note that a rank-$k$ maxout unit can identify $k$ cones from its input-domain, whereby each cone is a neighborhood of the positive half-ray $\{r\mathbf{W}_i \in \mathbb{R}^n : r \in \mathbb{R}_+\}$ corresponding to the gradient $\mathbf{W}_i$ of the linear function $f_i$ for all $i \in [k]$. Elaborating this observation, we obtain:

**Theorem 9.** *A maxout network with $L$ layers of width $n_0$ and rank $k$ can compute functions with at least $k^{L-1}k^{n_0}$ linear regions.*

Theorem 9 and Proposition 8 show that deep maxout networks can compute functions with a number of linear regions that grows exponentially with the number of layers, and exponentially faster than the maximal number of regions of shallow models with the same number of units. Similarly to the rectifier model, this exponential behavior can also be established with respect to the number of network parameters. We note that although certain functions that can be computed by maxout layers can also be computed by rectifier layers, the rectifier construction from last section leads to functions that are not computable by maxout networks (except in the rank-2 case). The proof of Theorem 9 is based on the same general arguments from Sec. 2, but uses a different construction than Theorem 5 (details in the Supplementary Material).

## 5 Conclusions and Outlook

We studied the complexity of functions computable by deep feedforward neural networks in terms of their number of linear regions. We specifically focused on deep neural networks having piecewise linear hidden units which have been found to provide superior performance in many machine learning applications recently. We discussed the idea that each layer of a deep model is able to identify pieces of its input in such a way that the composition of layers identifies an exponential number of input regions. This results in exponentially replicating the complexity of the functions computed in the higher layers. The functions computed in this way by deep models are complicated, but still they have an intrinsic rigidity caused by the replications, which may help deep models generalize to unseen samples better than shallow models.

This framework is applicable to any neural network that has a piecewise linear activation function. For example, if we consider a convolutional network with rectifier units, as the one used in (Krizhevsky et al. 2012), we can see that the convolution followed by max pooling at each layer identifies all patches of the input within a pooling region. This will let such a deep convolutional neural network recursively identify patches of the images of lower layers, resulting in exponentially many linear regions of the input space.

The structure of the linear regions depends on the type of units, e.g., hyperplane arrangements for shallow rectifier vs. Voronoi diagrams for shallow maxout networks. The pros and cons of each type of constraint will likely depend on the task and are not easily quantifiable at this point. As for the number of regions, in both maxout and rectifier networks we obtain an exponential increase with depth. However, our bounds are not conclusive about which model is more powerful in this respect. This is an interesting question that would be worth investigating in more detail.

The parameter space of a given network is partitioned into the regions where the resulting functions have corresponding linear regions. The combinatorics of such structures is in general hard to compute, even for simple hyperplane arrangements. One interesting question for future analysis is whether many regions of the parameter space of a given network correspond to functions which have a given number of linear regions.

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

## Footnotes

[1] A single sum-product hidden layer summarizes a layer of product units followed by a layer of sum units.
