[Supplementary Material]

# Supplementary Material:
# On the Number of Linear Regions of
# Deep Neural Networks

**Guido Montúfar**
Max Planck Institute for Mathematics in the Sciences
montufar@mis.mpg.de

**Razvan Pascanu**
Université de Montréal
pascanur@iro.umontreal.ca

**Kyunghyun Cho**
Université de Montréal
kyunghyun.cho@umontreal.ca

**Yoshua Bengio**
Université de Montréal, CIFAR Fellow
yoshua.bengio@umontreal.ca

## 6   Identification of Input-Space Neighborhoods

*Proof of Lemma 3.* Each output-space neighborhood $R \in P^L$ has as preimages all input-space neighborhoods that are $R$-identified by $\eta_L$ (i.e., the input-space neighborhoods whose image by $\eta_L$, the function computed by the first $L$-layers of the network, equals $R$). The number of input-space preimages of $R$ is denoted $\mathcal{N}_R^L$. If each $R \in P^L$ is the image of a distinct linear region of the function $h_L = g_L \circ f_L$ computed by the last layer, then, by continuity, all preimages of all different $R \in P^L$ belong to different linear regions of $\eta_L$. Therefore, the number of linear regions of functions computed by the entire network is at least equal to the sum of the number of preimages of all $R \in P^L$, which is just $\mathcal{N} = \sum_{R \in P^L} \mathcal{N}_R^{L-1}$.   □

## 7   Rectifier Networks

*Proof of Theorem 5.* The proof is done by counting the number of regions for a suitable choice of network parameters. The idea of the construction is to divide the first $L-1$ layers of the network into $n_0$ independent parts; one part for each input neuron. The parameters of each part are chosen in such a way that it folds its one-dimensional input-space many times into itself. For each part, the number of foldings per layer is equal to the number of units per layer. See Fig. 5.

As outlined above, we organize the units of each layer into $n_0$ non-empty groups of units of sizes $p_1, \ldots, p_{n_0}$. A simple choice of these sizes is, for example, $p_1 = \cdots = p_{n_0} = \lfloor n_l/n_0 \rfloor$ for a layer of width $n_l$, dropping the remainder units. We define the input weights of each group in such a way that the units in that group are sensitive to only one coordinate of the $n_0$-dimensional input-space. By the discussion from Sec. 3.1, choosing the input and bias weights in the right way, the alternating sum of the activations of the $p$ units within one group folds their input-space coordinate $p$ times into itself. Since the alternating sum of activations is an affine map, it can be absorbed in the preactivation function of the next layer. In order to make the arguments more transparent, we view the alternating sum $\tilde{h} = h_1 - h_2 + \cdots \pm h_p$ of the activations of the $p$ units in a group as the activation of a fictitious intermediary unit. The compound output of all these $n_0$ intermediary units partitions the input-space, $\mathbb{R}^{n_0}$, into a grid of $\prod_{i=1}^{n_0} p_i$ identified regions. Each of these input-space regions is mapped to the $n_0$-dimensional unit cube in the output-space of the intermediary layer. We view this unit cube as the *effective* inputs for the next hidden layer, and repeat the construction. In this way, with each layer of the network, the number of identified regions is multiplied by $\prod_{i=1}^{n_0} p_i$, according to Lemma 3. In the following we discuss the folding details explicitly.

Consider one of the network parts and consider the weights used in Sec. 3.1. The function $\tilde{h}$ computes an alternating sum of the responses $h_k$ for $k \in [p]$. It is sufficient to show that this sum folds the

Figure 5: Left: Illustration of the proof of Theorem 5. The figure shows a rectifier network that has been divided into $n_0$ independent parts. Each part is sensitive only to one coordinate of the input-space. Each layer of each part is fed to a fictitious intermediary affine unit (computed by the preactivation function of the next layer), which computes the activation value that is passed to the next layer. Right: Illustration of a function computed by the depicted rectifier network for $n_0 = 2$ at the intermediary layer. The function is composed of two foldings; the first pair of hidden units fold the input-space $\mathbb{R}^2$ along a line parallel to the x-axis, and the second pair, along a line parallel to the y-axis.

input-coordinate $p$ times into the interval $(0,1)$. Inspecting the values of $h_k$, we see that we only need to explore the intervals $(0,1),(1,2),\dots,(p-1,p)$.

Consider one of these intervals, $(k-1,k)$, for some $k \in [p]$. Then, for all $x \in (k-1,k)$, we have $\tilde{h}(x) = x + 2\sum_{i=1}^{k-1}(-1)^i(x-i) = (-1)^{k-1}(x-(k-1)) - \frac{1}{2}((-1)^{k-1}-1)$. Hence $\tilde{h}(k-1) = -\frac{1}{2}((-1)^{k-1}-1)$ and $\tilde{h}(k) = (-1)^{k-1} - \frac{1}{2}((-1)^{k-1}-1)$. One of the two values is always zero and the other one, and so $\tilde{h}(\{k-1,k\}) = \{0,1\}$. Since the function is linear between $k-1$ and $k$, we obtain that $\tilde{h}$ maps the interval $(k-1,k)$ to the interval $(0,1)$.

In total, the number of input-space neighborhoods that are mapped by the first $L-1$ layers onto the (open) unit hypercube $(0,1)^{n_0}$ of the (effective) output space of the $(L-1)$-th layer is given by

$$\mathcal{N}_{(0,1)^{n_0}}^{L-1} = \prod_{l=1}^{L-1}\prod_{i=1}^{n_0} p_{l,i}, \tag{5}$$

where $p_{l,i}$ is the number of units in the $i$-th group of units in the $l$-th layer.

The inputs and bias of the last hidden layer can be chosen in such a way that the function $h_L$ partitions its (effective) input neighborhood $(0,1)^{n_0}$ by an arrangement of $n_L$ hyperplanes in general position, i.e., into $\sum_{j=0}^{n_0}\binom{n_L}{j}$ regions (see Sec. 2.2).

Let $m_l$ denote the remainder of $n_l$ divided by $n_0$. Choosing $p_{l,1} = \cdots = p_{l,n_0-m_l} = \lfloor n_l/n_0 \rfloor$ and $p_{l,n_0-m_l+1} = \cdots = p_{l,n_0} = \lfloor n_l/n_0 \rfloor + 1$, we obtain a total of linear regions

$$\left(\prod_{l=1}^{L-1}\left\lfloor\frac{n_l}{n_0}\right\rfloor^{n_0-m_l}\left(\left\lfloor\frac{n_l}{n_0}\right\rfloor+1\right)^{m_l}\right)\sum_{j=0}^{n_0}\binom{n_L}{j}. \tag{6}$$

This is equal to the bound given in theorem when all remainders $m_l = n_l - n_0\lfloor n_l/n_0 \rfloor$ are zero, and otherwise it is larger. This completes the proof. $\quad\square$

## 8 Our Bounds in terms of Parameters

We computed bounds for the maximal number of linear regions of the functions computable by different networks in terms of their number of hidden units. It is not difficult to express these results in terms of

the number of parameters of the networks and to derive expressions for the asymptotic rate of growth of the number of linear regions per added parameter. This kind of expansions have been computed in Pascanu et al. (2013; Proposition 8). The number of parameters of a deep model with $L$ layers of width $n$ behaves as $\Theta(Ln^2)$, i.e., it is bounded above and below by $Ln^2$, asymptotically. The number of parameters of a shallow model with $Ln$ hidden units behaves as $\Theta(Ln)$. Our Theorem 5 and the discussion of shallow networks given in Sec. 2.2, imply the following asymptotic rates (number of linear regions per parameter):

- For a deep model: $\Omega\left(\left(n/n_0\right)^{n_0(L-1)} \frac{n^{n_0-2}}{L}\right)$.

- For a shallow model: $O\left(L^{n_0-1}n^{n_0-1}\right)$.

This shows that, for deep models, the maximal number of linear regions grows exponentially fast with the number of parameters, whereas, for shallow models, it grows only polynomially fast with the number of parameters.

# 9 Maxout Networks

*Proof of Proposition 8.* Here we investigate the maximal number of linear regions of a rank-$k$ maxout layer with $n$ inputs and $m$ outputs. In the case of rectifier units, the solutions is simply the maximal number of regions of a hyperplane arrangement. In the case of maxout units, we do not have hyperplane arrangements. However, we can upper bound the number of linear regions of a maxout layer by the number of regions of a hyperplane arrangement. The arguments are as follows.

As mentioned in Sec. 4, each maxout unit divides its input into the linear regions of an upper envelope of $k$ real valued linear functions. In other words, the input space is divided by pieces of hyperplanes defining the boundaries between inputs where one entry of the preactivation vector is larger than another. There are at most $k^2$ such boundaries, since each of them corresponds to the solution set of an equation of the form $f_i(\mathbf{x}) = f_j(\mathbf{x})$. If we extend each such boundary to a hyperplane, then the number of regions can only go up.

The linear regions of the layer are given by the intersections of the regions of the individual units. Hence, the number of linear regions of the layer is upper bounded (very loosely) by the number of regions of an arrangement of $k^2 \cdot m$ hyperplanes in $n$-dimensional space. By Zaslavsky (1975), the latter is $\sum_{j=0}^{n} \binom{k^2 m}{j}$, which behaves as $O((k^2 m)^n)$, i.e., polynomially in $k$ and in $m$. □

*Proof of Theorem 9.* Consider a network with $n = n_0$ maxout units of rank $k$ in each layer. See Fig. 6. We define the seeds of the maxout unit $q_j$ such that $\{\mathbf{W}_{i,:}\}_i$ are unit vectors pointing in the positive and negative direction of $\lfloor k/2 \rfloor$ coordinate vectors. If $k$ is larger than $2n_0$, then we forget about $k - 2n_0$ of them (just choose $\mathbf{W}_{i,:} = 0$ for $i > 2n_0$). In this case, $q_j$ is symmetric about the coordinate hyperplanes with normals $e_i$ with $i \leq \lfloor k/2 \rfloor$ and has one linear region for each such $i$, with gradient $e_i$. For the remaining $q_j$ we consider similar functions, whereby we change the coordinate system by a slight rotation in some independent direction.

This implies that the output of each $q_j \circ (f_1, \ldots, f_k)$ is an interval $[0, \infty)$. The linear regions of each such composition divide the input space into $r$ regions $R_{j,1}, \ldots, R_{j,k}$. Since the change of coordinates used for each of them is a slight rotation in independent directions, we have that $R_i := \cap_j R_{j,i}$ is a cone of dimension $n_0$ for all $i \in [k]$. Furthermore, the gradients of $q_j \circ f_j$ for $j \in [n_0]$ on each $R_i$ are a basis of $\mathbb{R}^{n_0}$. Hence the image of each $R_i$ by the maxout layer contains an open cone of $\mathbb{R}^{n_0}$ which is identical for all $i \in [k]$. This image can be shifted by bias terms such that the effective input of the next layer contains an open neighborhood of the origin of $\mathbb{R}^{n_0}$.

The above arguments show that a maxout layer of width $n_0$ and rank $k$ can identify at least $k$ regions of its input. A network with $L - 1$ layers with therefore identify $k^{L-1}$ regions of the input. □

In Sec. 4 we mentioned that maxout layers can compute functions whose linear regions correspond to intersections of Voronoi diagrams. Describing intersections of Voronoi diagrams is difficult, in general. There are some superpositions of Voronoi diagrams that correspond to hyperplane arrangements which are well understood. Here are two particularly nice examples:

Figure 6: Left: Illustration of a rank-2 maxout layer with $n = 3$ inputs and $m = 2$ outputs. The preactivation function maps the input into $mk$-dimensional space, where $k = 2$ is the rank of the layer. The activation function maximizes over groups of $k$ preactivation values. Right: Illustration of the 3-dimensional Shi arrangement $\mathcal{S}_3 \subset \mathbb{R}^3$ (depicted is the intersection with $\{\mathbf{x} \in \mathbb{R}^3 \colon \mathbf{x}_1 + \mathbf{x}_2 + \mathbf{x}_3 = 1\}$). This arrangement corresponds an input-space partition of a rank-3 maxout layer with $n = 3$ inputs and $m = 3$ outputs (for one choice of the parameters). Each pair of parallel hyperplanes delimits the linear regions of one maxout unit.

**Example 10.** Consider a layer with $n$ inputs and $m = n(n-1)/2$ rank-3 maxout units labeled by the pairs $(i, j)$, $1 \leq i < j \leq n$. The input and bias weights can be chosen in such a way that the regions of unit $(i, j)$ are delimited by the two hyperplanes $H_{(i,j),s} = \{\mathbf{x} \in \mathbb{R}^n \colon \mathbf{x}_i - \mathbf{x}_j = s\}$ for $s \in \{0, 1\}$. The intersections of the regions of all units are given by the regions of the hyperplane arrangement $\{H_{(i,j),s}\}_{1 \leq i < j \leq n, s = 1, 2}$, which is known as the *Shi arrangement* $\mathcal{S}_n$ and has $(n + 1)^{n-1}$ regions. The right panel of Fig. 6 illustrates the Shi arrangement $\mathcal{S}_3$.

**Example 11.** A related arrangement, corresponding to rank-4 maxout units, is the *Catalan arrangement*, which has triplets of parallel hyperplanes, and a total of $n! C_n$ regions, where $C_n := \frac{1}{n+1}\binom{2n}{n}$ is the *Catalan number*. For details on these arrangements see (Stanley 2004; Corollary 5.1 and Proposition 5.15).

## 10  Other Networks

In the introduction we mention that our analysis of rectifier and maxout networks serves as a platform to study other types of feedforward neural networks. Without going into many details, we exemplify this for the particular case of convolutional networks. A convolutional network is a network whose units take values in a space of *features* (real valued arrays) and whose edges pass features by convolution with *filters* (real valued arrays). Since convolution is a linear map, the preactivation function of a convolutional network is of the same form as the preactivation functions considered in this paper. Its output is a feature, but it can be written as a vector, like the $f_{l,i}$'s that we considered. Hence convolutional networks with piecewise linear activations fall in the class of networks that we considered here. The only difference lies in that the corresponding input weight matrices of convolutional networks belong to restricted classes of matrices.

## 11  Sinusoidal Boundary Experiment

Here, we describe the experiments we performed to obtain Fig. 1 in the main text.

In this experiment we considered two MLPs, of which one has a single hidden layer with 20 hidden units and the other has two hidden layers with 10 hidden unit each. The MLPs were trained on the same synthetic dataset using a conjugate natural gradient (Pascanu and Bengio 2014) which was used to minimize the effect of optimization. We plot the best of several runs. The shallow model misclassified 123 examples, whereas the deep model did only 24 examples. The two-layer model is better at capturing a sinusoidal decision boundary, because it can define more linear regions.

## 12 Visualizing the Behavior of Hidden Units in Higher Layers

In this section, we describe the details on how one can visualize the effect of folding in rectifier MLPs, discussed in Sec. 2.6.

Any piecewise linear function is fully defined by the different linear pieces from which it is composed. Each piece is given by its domain (a region of the input space $R_i \subseteq \mathbb{R}^{n_0}$) and the linear map $f_i$ that describes its behavior on $R_i$. Because $f_i$ is an affine map, it can be interpreted in the same way hidden units in a shallow model are. Namely, we can write $f_i$ as:

$$f_i(\mathbf{x}) = \mathbf{u}^\top \mathbf{x} + c, \quad \mathbf{x} \in R_i,$$

where $\mathbf{u}^\top$ is a row vector, and $\mathbf{u}^\top \in \mathbb{R}^{n_0}$. Then $f_i$ measures the (unnormalized) cosine distance between $\mathbf{x}$ and $\mathbf{u}^\top$. If $\mathbf{x}$ is some image, $\mathbf{u}^\top$ is also an image and shows the pattern (template) to which the unit responds whenever $\mathbf{x} \in R_i$.

Given an input example $\mathbf{x}$ from an arbitrary region $R_i$ of the input space we can *construct* the corresponding linear map $f_i$ generated by the $j$-th unit at the $l$-th layer. Specifically, the weight $\mathbf{u}^\top$ of the linear map $f_i$ is computed by

$$\mathbf{u}^\top = (\mathbf{W}_l)_{j:} \operatorname{diag}\left(\mathbf{I}_{f_{l-1}>0}(\mathbf{x})\right) \mathbf{W}_{l-1} \cdots \operatorname{diag}\left(\mathbf{I}_{f_1>0}(\mathbf{x})\right) \mathbf{W}_1. \tag{7}$$

The bias of the linear map can be computed in a similar way.

From Eq. (7), we see that the linear map of a specific hidden unit $f_{l,j}$ at the $l$-th layer can be found by keeping track of which linear piece is used at each layer until the $l$-th layer ($\mathbf{I}_{f_p>0}, p < l$, which is the indicator function). At the end, the $j$-th row $(\mathbf{W}_l)_{j:}$ of the weight matrix $\mathbf{W}_l$ is multiplied. Although we present a formula specific to the rectifier MLP, it is straightforward to adapt this to any MLP with piecewise linear activations (to convolutional neural networks with maxout activation, for example).

The linear map computed by Eq. (7) depends on the specific point $\mathbf{x}$, and so, we need to traverse a set of points (e.g., training samples) in order to identify different linear responses of a hidden unit. While this does not necessarily give all possible responses (say if there are less training samples than distinct response regions), if the set of points is large enough, we can get sufficiently many to provide a better understanding of its behavior.

We trained a rectifier MLP with three hidden layers on the Toronto Faces Dataset (TFD) (Susskind et al. 2010). The first two hidden layers having 1000 hidden units each and the last one 100. We trained the model using stochastic gradient descent. We used, as regularization, an $L_2$ penalty with a coefficient of $10^{-3}$, dropout on the first two hidden layers (with a drop probability of $0.5$) and we enforced the weights to have unit norm column-wise by projecting the weights after each SGD step. We used a learning rate of $0.1$ and an output layer composed of sigmoid units. The purpose of these regularization schemes, and the sigmoid output layer is to obtain cleaner and sharper filters. The model was trained on fold 1 of the dataset and achieved an error of 20.49% which is reasonable for this dataset and a non-convolutional model.

Since each unit in the first hidden layer only responds to one linear region, we directly visualize the learned weight vectors of 16 randomly selected units in that layer. These are shown on the top row of Fig. 7. For each other hidden layer, we randomly pick 20 units and visualize the most interesting 4, based on the maximal Euclidean distance between the different linear responses of each unit. The linear responses of each unit are computed by clustering the responses obtained on the training set (we only consider those responses where the activation was positive) into four clusters using K-means algorithm. We show the representative linear response in each of the clusters (see the second and third rows of Fig. 7). Similarly, we visualize the linear maps learned by each of the output units. Fig. 8 shows a visualization for all seven units in the output layer.

By looking at the differences among the distinct linear regions that a hidden unit responds to, we can investigate the type of invariance the unit learned. In Fig. 9, we show the differences among the four linear maps learned by the last visualized hidden unit of the third hidden layer (the last column of the visualized linear maps). From these visualizations, we can see that a hidden unit learns to be invariant to more abstract and interesting translations at higher layers. We also see the types of invariance of a hidden unit in a higher layer clearly.

Zeiler and Fergus (2013) attempt to visualize the behavior of units in the upper layer, specifically, of a deep convolutional network with rectifiers. This approach is to some extent similar to our approach proposed here, except that we do not make any other assumption beside that a hidden unit in a networks

Figure 7: Visualizations of the linear maps learned by a selection of units in each hidden layer of a rectifier MLP trained on the TFD dataset. Each row block corresponds to one hidden layer. The first block column shows the unnormalized linear maps, and the second block column shows the normalized linear map (shown is only the direction of each map). For the first layer, shown are the maps of 16 units. For the second and third layers, shown are 4 maps for each of 4 units (each column corresponds to one unit). Colors are only used to improve the distinction among different filters.

uses a piece-wise linear activation function. The perspective from which the visualization is considered is also different. Zeiler and Fergus (2013) approaches the problem of visualization from the perspective of (approximately) inverting the feedforward computation of the neural network, whereas our approach is derived by identifying a set of linear maps per hidden unit. This difference leads to a number of minor differences in the actual implementation. For instance, Zeiler and Fergus (2013) approximates the inverse of a rectifier by simply using another rectifier. On the other hand, we do not need to approximate the inverse of the rectifier. Rather, we try to identify regions in the input space that maps to the same activation.

In our approach, it is possible to visualize an actual point in the input space that maps to the same activation of a hidden unit. In Fig. 10, we show three distinct points in the input space that activate a randomly chosen hidden unit in the third hidden layer to be exactly $2.5$. We found these points by first finding three training samples that map to an activation close to $2.5$ of the same hidden unit, and from each found sample, we search along the linear map (computed by Eq. (7)) for a point that exactly results in the activation of $2.5$. Obviously, the found point is *not* one of the training samples. From those three points, we can see that the chosen hidden unit responds to a face with wide-open mouth and a set of open eyes while being invariant to other features of a face (e.g., eye brows). By the perturbation analysis, we can assume that there is an open set around each of these points that are identified by the hidden unit.

Linear maps at output layer          Normalized

7 units × 4 maps

Figure 8: Linear maps of the output units of the rectifier MLP trained on TFD dataset. The corresponding class labels for the columns are (1) anger, (2) disgust, (3) fear, (4) happy, (5) sad, (6) surprise and (7) neutral.

Difference          Normalized

Figure 9: Differences between four linear regions of one hidden unit at the third hidden layer of the rectifier MLP trained on TFD.

Figure 10: Visualization of three distinct points in the input space that map to the same activation of a randomly chosen unit at the third hidden layer. The top row shows three points (*not* training/test samples) in the input space, and for each point, we plot the linear map below.