[Reviews · NeurIPS 2014]

Submitted by Assigned_Reviewer_35

I. Introduction
1) Limited number of papers have investigated deep networks from a theoretical perspective?? I disagree with this statement, many papers from Bengio's group do just this.
2) Define what you mean by popular networks, this is too vague (is it CNNs, RNNs, LSTMs, different non-linearities)
3) Pascanu did the analysis for ReLU, pls justify the added benefit for doing this for piecewise linear functions? How novel is your work compared to Pascanu-13, as he did this for rectified units. You should be more clear about your novelties compared to this paper.
4) Pls quantify what significant improvement means??

II. Feedforward Neural Networks
a) It woudl be helpful to have some pictures to go along with 2.2, as Pascanu-13 gives, or include this in supplemental material and point to it
b) does the idea of input space foldings also depend on the non-linearity or no? this should be stated more clearly?
c) So what is the bound on \epsilon to guarantee it stable to perturbation, pls clarify?
d) There is very little connection between sections. It would be better if the flow between sub-sections was better.

III. Deep Rectifier Networks
a) it would be helpful if you had a visual description for 3.1 in the paper (or supplemental material)
b) provide a deeper analysis of what the theorem does

IV. Deep Maxout Networks
a) again, pictures for this section would be helpful as well
b) a better comparison and analysis for linear regions for rectified and maxout would help the readers understand why one could be preferred over another. This is a weakness of the paper

V. Conclusions
a) How do the proofs in Theorem 4 and 8 help to show that deep models generalize better to unseen examples
Summary: 2 line summary:
While this paper presents an interesting analysis of the number of linear regions of neural networks, it can be improved in two main areas
1) providing more intuitive visual pictures that help the readers to understand the math
2) a better comparison between linear regions for Relu and maxout

Submitted by Assigned_Reviewer_41

This paper proposes a construction that shows how deep networks of rectifier or maxout units lead to combinatorially many more linear regions (as input space pre-image) than shallow networks; compared to earlier work, this construction provides a new, better lower bound on the maximal number of linear regions for deep networks.

The paper gives some intuitive understanding of what this linear regions in input space would be, with graphical illustrations that show how symmetries in the encoding function (e.g., absolute value) lead to folding of the input space and cristal-like replication of regions in input space that will land into the same linear output region.
Then explicit constructions are provided for maxout and rectifier units.

These points are not intuitively surprising but it's nice to have explicit bounds, the paper is clear enough. However the contribution seems rather narrow, perhaps too much so.
Summary: This paper proposes a new lower bound on the maximal number of linear regions created by deep networks of maxout or rectifier units, and shows that it grows exponentially compared to shallow networks. This is interesting work but the contribution may be too narrow.

Submitted by Assigned_Reviewer_46

This paper presents a theoretical analysis of deep neural networks whose nonlinear activation functions are given by the rectifier function or maxout units. The work builds on and generalizes earlier work by Pascanu et al. (2013) and demonstrates that the number of linear regions in a deep rectifier network grows exponentially in the number of layers and polynomially in the size of the layer, which is much faster than comparable shallow networks. A similar analysis is performed for maxout networks.

Overall, the paper is well written and there are number of useful metaphors introduced in the analysis. I like the use of the folding metaphor to demonstrate the compositional properties of the networks. The visualization of the network provided in the supplemental material was also useful to ground some of the theory with empirical findings.

The paper is fairly clear, though some of the sections took several readings to understand. But I'm not sure the clarity could be improved without significant additional space.

One question that remains for me centers on the stability to perturbation. While the analysis on Section 2.5 studies a perturbation in the model parameters, can the same argument be made for a perturbation in the input? This seems important in light of the folding metaphor and the compositionality where different regions are mapped to the same higher level representation. It would be nice to have a theoretical result that depth improves invariance in the same way as has been shown for deep networks with sigmoid units.
Summary: This paper presents an analysis of deep rectifier networks that shows the number of linear regions in a deep rectifier network grows exponentially in the number of layers and polynomially in the size of the layer, which is much faster than comparable shallow networks. A similar analysis is performed for maxout networks.
Author Feedback
Author rebuttal: Dear reviewers,

We would like to thank you for the thorough and insightful comments.

=R35=

*I-1
We are aware of the theoretical work from Bengio's lab. What we meant is that, despite those earlier contributions, we feel that many questions remain open. Much of the available theoretical work is either too narrow (showing the efficiency of deep networks for very restricted classes of functions) or focusses on models that are not as widely used in practice. Our work looks at a family of models that is presently widely used in industry and academia. Furthermore, it shows the benefits of depth for the representation of a rather large family of functions (with certain types of symmetries or invariances).

*I-2
We meant feedforward networks with piece-wise linear activations (rectifier and maxout units). We will make it clearer in the final version. We think that our proofs are easily extensible to convolutional networks with rectifier and maxout units.

*I-3
We will clarify more carefully the novelty of our work. In short, we present more general and accurate results than Pascanu et al. did, offering a significant improvement over their lower bounds, and upper bounds. We provide an analysis of maxout networks, and worked out a formal description of the main mechanism behind the proofs; the identification of input regions and input space folding. From the idea of space folding we understand the families of functions that can be represented more efficiently by deep models (those with compounded symmetries or invariances).

*I-4
Our lower bound is a significant improvement over that by Pascanu et al., as it demonstrates the advantages of depth even for moderate depth, whereas they demonstrated only for very deep models. Their result has an exponent L-1, while we have an exponent (L-1)n_0. L is the number of layers and n_0 is the number of input neurons.

*II-a
We will add such a figure in the supplementary material.

*II-b
Space folding in our definition means that the model (or some intermediary hidden units) has the same response for different inputs. This describes a form of non-injective behaviour, which can be used to study any function. The specific way in which the space is folded, however, depends on the type and the composition of nonlinearities. Although the methods presented here focuses on combinatorial properties to piece-wise linear activations, it may be possible to extend it to differentiable nonlinear functions using topology.

*II-c
The intention of our perturbation analysis is merely to show that there are generic choices of parameters for which the functions computed by the network have at least as many regions as specified by our bounds, i.e., our bounds are not a "0-probability event". Although it is hard to provide a bound on epsilon, under natural assumptions, we think the bound shrinks exponentially with depth. Consider, e.g., how much one has to perturb N lines on a plane until two of them are parallel. For some parametrization and some metric on the parameter space, for large N, epsilon will become arbitrarily small. On the other hand, generically, no two of N lines are parallel. We are working on providing a better quantifiable measure of this analysis.

*II-d
Thank you for the suggestion. We will improve the text flow.

*III-a
Figs. 4 and 1 in the supplementary material are intended to illustrate 3.1. We are not sure how to improve its visual description.

*III-b
We will improve the discussion and provide more explanations.

*IV-a
We will add more illustrations in the supplementary material.

*IV-b
Thanks for the suggestion. The linear regions of rectifier and maxout networks have a different structure (hyperplane arrangements vs. Voronoi diagrams). The tradeoff between these constraints will likely depend on the task and is not easily quantifiable at this point. As for the number of regions, in both cases we obtain an exponential increase with depth; however, our bounds are not conclusive about which model is more powerful in this respect. It is an interesting question that is worth investigating in more detail. We will add comments in the Conclusion.

*V-a
These theorems do not directly imply generalization performance. However, they show that deep networks use few parameters to represent function with many regions. This means that learning a few of the regions fixes the other regions. Therefore deep models can better represent functions that have lots of symmetries (or compounded invariances) via a proper space folding. Within this class of functions we expect them to generalize well to unseen samples.

=R41=

*"contribution seems rather narrow"
Please see Answer 1 to R35. Our results are not narrow, because they apply to a wide variety of models that are used both in industry and academia. They apply to a larger class of functions that present a multitude of compounded symmetries or invariances, represented by sequential foldings of space. Other theoretical work mostly focus on very specific class of functions or less popular models. Based on the folding idea, we present an intuitive picture of the structures where deep models can generalize well. We believe that our analysis can be used for studying other interesting properties of deep models.

=R46=

*".. perturbation in the input?"
Yes. Exactly the same argument will apply to perturbations in the input. We wish to purse the implications for invariance and the classes of them that can be learnt.

*".. with sigmoid units."
We are not aware of theoretical work showing that depth improves invariance in networks with sigmoid units. The analysis that we present is more polyhedral-combinatorial in nature and does not apply directly to sigmoid units. However, the folding perspective still applies. Since for sigmoid units there are no linear regions, it will require another approach focusing the cardinality of the pre-images of the outputs.